# QLASS: Boosting Language Agent Inference via Q-Guided Stepwise Search

**Zongyu Lin** [1] [*]   **Yao Tang** [2] [*]   **Xingcheng Yao** [1] [*]   **Da Yin** [1] [*]   **Ziniu Hu** [1]   **Yizhou Sun** [1] [†]   **Kai-Wei Chang** [1] [†]

## Abstract

Language agents have become a promising solution to complex interactive tasks. One of the key ingredients to the success of language agents is the reward model on the trajectory of the agentic workflow, which provides valuable guidance during training or inference. However, due to the lack of annotations of intermediate interactions, most existing works use an outcome reward model to optimize policies across entire trajectories. This may lead to sub-optimal policies and hinder the overall performance. To address this, we propose **QLASS** (**Q**-guided **L**anguage **A**gent **S**tepwise **S**earch), to automatically generate annotations by estimating Q-values in a stepwise manner for open language agents. By introducing a exploration tree and performing process reward modeling, QLASS provides effective intermediate guidance for each step. With the stepwise guidance, we propose a Q-guided generation strategy to enable language agents to better adapt to long-term value, resulting in significant performance improvement during model inference on complex interactive agent tasks. Notably, even with almost half the annotated data, QLASS retains strong performance, demonstrating its efficiency in handling limited supervision. We also empirically show that QLASS can lead to more effective decision making through qualitative analysis. [1]

## 1. Introduction

Supervised fine-tuning (SFT) is commonly employed to make base LLMs perform effective reasoning and planning in complex agent tasks by imitating expert trajectories (Chen

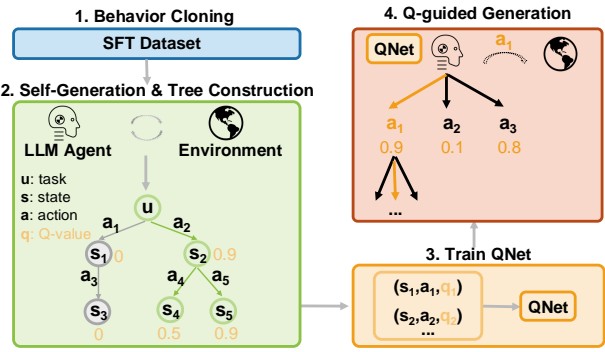

Figure 1. QLASS pipeline overview. QLASS involves mainly four stages: **1)** Supervised fine-tuning (SFT) on expert data. **2)** Leverage SFT agent to explore the environment and construct an exploration tree for each task. After construction, estimate the Q-value of each tree node based on Equation 7. **3)** Train QNet on the estimated Q-values. **4)** Use the trained QNet to provide inference guidance at each step.

et al., 2023; Yin et al., 2024). However, the substantial human annotations required to collect training data present a significant bottleneck, limiting both performance and scalability. This challenge is particularly pronounced in agent tasks (Yao et al., 2022; Shridhar et al., 2021; Wang et al., 2022a), where data scarcity is a critical issue due to the inherent complexity and diversity of real-world interactions. To overcome this challenge, self-improvement techniques have shown to be a promising area of research, enabling LLMs to learn from self-generated data without extensive human intervention (Wang et al., 2022b; Singh et al., 2023; Hosseini et al., 2024; Zhang et al., 2024), and most recently, enable LLMs to improve their outputs by scaling up their test-time computation in a more intelligent way (Wang et al., 2024; Shinn et al., 2023; Snell et al., 2024). Inspired by this, we focus on improving the inference-time search for language agents, which is a crucial technique for the success of more generalizable agents in real-world environments.

An essential component of inference-time scaling methods is the reward model (Snell et al., 2024), which evaluates the quality of self-explored data. Many existing works derive a single outcome reward based on ground-truth (Wang et al., 2024; Shinn et al., 2023). Although this approach is straightforward, it falls short in handling complex agent tasks, since an outcome-reward model cannot accurately score each step within a long trajectory in intricate scenarios. In addition, a

*Equal contribution. † Equal advising. [1]University of California, Los Angeles, USA [2]Shanghai Jiaotong University, Shanghai, China. Correspondence to: Yizhou Sun <yizhou.sun@ucla.edu>, Kai-Wei Chang <kaiwei.chang@ucla.edu>.

*Proceedings of the $42^{nd}$ International Conference on Machine Learning*, Vancouver, Canada. PMLR 267, 2025. Copyright 2025 by the author(s).

[1]We will release our code and data in https://github.com/Rafa-zy/QLASS

trajectory achieving a high final outcome reward does not necessarily indicate that every action taken was optimal; the agent may have completed the task successfully, but some actions could have been inefficient or suboptimal (Uesato et al., 2022).

Therefore, a good process reward model is necessary to learn from the environmental feedback and provide stepwise evaluations of agent actions. This model enables the agent to fully understand and learn from the intermediate stages of complex tasks, ultimately improving performance and generalization. The key challenge lies in developing an effective process reward model for self-improvement without relying on extensive human annotations for the stepwise reward. There has been a thread of work that has focused on process reward modeling (Uesato et al., 2022; Lightman et al., 2023; Wang et al., 2023; Chen et al., 2024). However, these methods rely on either costly human annotation or computationally heavy random rollouts, rendering them inefficient for the self-improvement of language model agents.

To reduce the annotation reliance on process rewards, we propose QLASS to perform effective process reward modeling to guide agent inference. Specifically, we explicitly formalize the self-generated exploratory trajectories as exploration trees and update the process rewards of all the tree nodes based on the tree structures. To better capture the future utility at each step of a multi-turn reasoning process, we employ the Bellman equation (Bellman & Dreyfus, 2015) to learn a Q-based process reward. Unlike simple outcome-based rewards, this Q-value captures how immediate decisions contribute to longer-term payoffs, enabling finer-grained control over the reasoning trajectory. Moreover, the Bellman update rule iteratively refines the estimated Q-values by propagating future rewards back to earlier states, reducing reliance on sparse or delayed feedback signals. This allows us to efficiently gather supervision for state-action pairs without requiring explicit annotations of full trajectories. With these Q values in hand, we can then train a function approximator (QNet) (Watkins & Dayan, 1992) to predict the expected return of any partial solution, ultimately providing a strong inductive bias to guide open-language agents. By prioritizing actions with higher estimated Q values, the agent steers its own reasoning in a more targeted manner, facilitating efficient stepwise planning within expansive search spaces.

While recent attempts like KIMI-k1.5 (Team et al., 2025) and Deepseek-R1 (Guo et al., 2025) report failures in process reward modeling, we argue that such modeling is indispensable for agent tasks. Back-and-forth agent behaviors inherently create stepwise inefficiencies (e.g., repetitive environment queries or cyclic reasoning), which the sparse outcome rewards cannot diagnose. Our Q-value estimation directly addresses this by propagating future utility backward through trajectories, dynamically pruning actions with low reward while preserving critical decision points. This enables agents to disentangle productive reasoning from wasteful loops, even with limited supervision. To summarize, our contribution can be divided into three folds:

**1) Process Reward Modeling with Q-Value Estimation**: We introduce QLASS, a novel strategy that leverages estimated Q-values to generate intermediate annotations for language agents, providing stepwise guidance for model inference. We visualize the overall framework of QLASS in Figure 1.

**2) Q-Guided Generation Strategy**: We propose a Q-guided generation technique that significantly enhances agent performance via process-based guidance during inference, ensuring effective decision making at each step.

**3) Superior Performance with Limited Supervision**: QLASS shows strong performance on a set of diverse agent environments, including WebShop, ALFWorld, and SciWorld. QLASS can give effective inference-time guidance even when nearly half of the annotated data is reduced. These experimental results highlight the efficiency and robustness of QLASS in scenarios with limited supervision.

## 2. Related Work

### 2.1. Large Language Model Agent

Large language models have shown impressive performance in complex interactive tasks, such as web navigation (Yao et al., 2022), scientific reasoning (Wang et al., 2022a), and action planning in embodied environments (Shridhar et al., 2021). ReAct (Yao et al., 2023) developed a prompting method to shape language models as agents that can reason and act. While several works (Shen et al., 2024; Song et al., 2023) improve agent performance with closed-source LLM controllers, the open-source LLM agents still offer unique advantages like accessibility and customization. FireAct (Chen et al., 2023) and LUMOS (Yin et al., 2024) leverage high-quality data generated by experts and employ teacher-forcing to improve the performance of open-source agents. Koh et al. (2024) uses best-first tree search for language agents with the value function given by GPT-4o. In line with this, our QLASS is also based on open-source LLMs.

### 2.2. Self-Improvement for LLM

The self improvement of LLM can be a good way to improve LLM without heavy human annotation, which can be divided into two parts. (1) Training models on self-generated data is a promising approach. A large number of works (Dou et al., 2024; Wang et al., 2022b; Yuan et al., 2023; Singh et al., 2023; Gulcehre et al., 2023; Wang et al., 2023) follow the paradigm of self-training, which filters

positive self-generated data and performs model training on those filtered data. Some other works (Song et al., 2024; Setlur et al., 2024) utilize both positive and negative data to construct preference pairs and update the policy using direct preference optimization (Rafailov et al., 2024). (2) Another approach is to scale up the computation of inference to improve the outputs of LLMs. The methods include guiding the inference based on scalar-based reward models (Wang et al., 2024; Xu et al., 2022; Zhai et al., 2024) and modifying the output conditioning on the language feedback (critique provided by the LLM itself or another critique LLM) (Zhou et al., 2024; Wu et al., 2024; Shinn et al., 2023). In our paper, we focus on the self-improvement at inference time using our proposed process reward models.

### 2.3. Process Reward Modeling for LLM

Existing works have explored various strategies and reasoning policies for process reward modeling. (Uesato et al., 2022) and (Lightman et al., 2023) utilize human-annotated step-level correctness to train a reward model. Math-Shepherd (Wang et al., 2023) infers per-step rewards through random rollouts. TS-LLM (Feng et al., 2023) employs an MCTS-based policy and infers per-step rewards using the TD-$\lambda$ (Sutton, 1988) method. ReST-MCTS* (Zhang et al., 2024) uses Monte Carlo tree search (MCTS) with re-inforced self-training to enhance the diversity and performance on general reasoning tasks like maths, science, and code. Most recently, Wang et al. (2024) Zhai et al. (2024) and Chen et al. (2025) also use step-level guidance for agent inference through training a step-level value model. Putta et al. (2024) applies a hybrid process reward modeling for web navigation tasks by combining Monte Carlo Tree Search (MCTS) rewards with scores generated by large language models to form process rewards. Our approach focuses on solving complex agent tasks by providing effective per-step guidance for LLM agent inference. Our method differs from Putta et al. (2024) because we do not rely on a strong proprietary LLM to provide rewards. Compared with Wang et al. (2024) and Zhai et al. (2024), we shift our focus on more complex agent tasks with larger search space and deeper search depth like ALFWorld and SciWorld. Compared with Zhang et al. (2024), our framework is much simpler with less stages of training and more straightforward to make process reward modeling works better compared with strong training-based baselines.

### 3. Preliminaries

In this section, we introduce Q-learning, the key algorithm that inspires QLASS to extract Q-values from the exploration trees. Q-learning (Watkins & Dayan, 1992) is a traditional model-free reinforcement learning algorithm, where a value function $Q(s, a)$ is trained to represent the expected

future rewards by taking action $a$ given state $s$. The optimal Q-function can be written as,

$$
Q^\star(s, a) = \max_\pi \mathbb{E}[r_t + \gamma r_{t+1} + \gamma^2 r_{t+2} + \cdots \mid \\ s_t = s, a_t = a, \pi], \tag{1}
$$

where $\pi$ is the policy, $\gamma$ is the discount factor, and $r_t$ is the received reward at step $t$. Given the definition of optimal Q-fucntion in Equation 1, the Bellman Optimality Equation (Bellman & Dreyfus, 2015) of Q-function can be written as,

$$
Q^\star(s_t, a_t) = r_t + \gamma \max_{a \in \mathcal{A}} Q^\star(s_{t+1}, a). \tag{2}
$$

In Q-learning, the value model $Q(s_t, a_t)$ is updated iteratively by,

$$
Q(s_t, a_t) \leftarrow (1 - \alpha) Q(s_t, a_t) \tag{3}
$$
$$
+ \alpha(r_t + \gamma \max_{a \in \mathcal{A}} Q(s_{t+1}, a)), \tag{4}
$$

where $\alpha$ is the learning rate and $\mathcal{A}$ is the action space. Combining immediate rewards from the current action and future potential rewards from subsequent actions, Q-value can be interpreted as the expected long-term value of taking a specific action in a given state.

In complex interactive tasks, the agent needs to account not only for immediate rewards but also for the potential long-term effects of its current decisions. This is where the Q-value becomes essential. However, directly adapting RL algorithms such as Q-learning to language agents can be sample-inefficient (Jin et al., 2018). This is because the action space in language agent tasks is typically a vast vocabulary, which may lead to an explosion of potential action sequences to be explored. To address this challenge, our approach successfully adapts Q-value extraction to language agent tasks by introducing an exploration tree, which we will introduce in the next section.

### 4. QLASS Pipeline Details

In this section, we will follow the order of QLASS pipeline and introduce each critical component step by step. The overall pipeline is shown in Figure 1 and Algorithm 1.

First, we will describe the initial stage of behavior cloning. Then, we will explain how the exploration tree is constructed during the second *self-generation* stage and how we use it to extract Q-values as the supervision to train Q-network (*QNet Training*). Finally, we will detail the *Q-guided generation* how the QNet is employed to guide the agent's test-time inference in a stepwise manner.

#### 4.1. Behavioral Cloning

Behavior cloning provides a strong initial foundation for language agents by supervised fine-tuning on expert trajec-

---

**Algorithm 1** General QLASS Pipeline

---

**Input:** Expert dataset $\mathcal{D}_{expert} = \{(u_i, a_t^i, o_t^i)_{t=1}^T\}_{i=1}^{N_e}$, policy $\pi_\theta$, QNet $\mathcal{Q}_\phi$
**Stage 1: Behavior Cloning**
Train $\pi_\theta$ on $\mathcal{D}_{expert}$ minimizing loss 5
**Stage 2: Construct Reasoning Trees**
**for** $i = 1$ **to** $N_e$ **do**
    Construct a reasoning tree with Algorithm 2
    Update Q-values recursively with Equation 7
**end for**
Collect Q-values from $\{T_i\}_{i=1}^N$ as dataset $\mathcal{D}_Q$
**Stage 3: QNet Training**
Train QNet $\mathcal{Q}_\phi$ on dataset $\mathcal{D}_Q$
**Step 4: Q-guided Generation**
Use QNet $\mathcal{Q}_\phi$ to score state-actions at each step

---

tories. Formally, the first stage of QLASS is to supervised fine-tune our language agent, denoted as the policy $\pi$, on a set of annotated samples $\mathcal{D}_{expert}$. We use ReAct (Yao et al., 2023)-style data for supervised fine-tuning, which additionally generates Chain-of-Thought (CoT) (Wei et al., 2022) reasoning paths before executing each action. We will use $a$ to denote the complete ReAct-style response generated by $\pi$ for simplicity.

Formally, given a dataset $\mathcal{D}_{expert} = \{(u_i, a_t^i, o_t^i)_{t=1}^T\}_{i=1}^{N_e}$, where $u_i$ represents the task description, $T$ is the trajectory length, $N_e$ is the number of trajectories in the expert dataset, $o_t^i$ is the environment observation after taking action $a_t^i$ at step $t$, we optimize the policy $\pi$ by minimizing the negative log-likelihood loss:

$$\mathcal{L}(\theta) = -\sum_i \sum_t \log \pi_\theta(a_t^i \mid u_i, a_{<t}^i, o_{<t}^i), \quad (5)$$

where $\theta$ denotes the parameters of the policy model $\pi_\theta$, which outputs the probability of action $a$ given task description $u$ and historical interactions $h_t = \{a_{<t}, o_{<t}\}$.

## 4.2. Constructing an Exploration Tree

The supervised fine-tuned agents can explore the environment and collect a large amount of trajectories. However, due to the extremely large search space of language agents, directly sampling trajectories without any guidance may lead to low efficiency. To address this issue, we propose to construct an exploration tree during ***self-generation***.

### 4.2.1. TREE STRUCTURE

For a trajectory, we take the task description as the root node, and each node below the root node is composed of the state, action, and related information for each step. For all trajectories of a task, they can be seen as different branches that originate from the same root node.

Specifically, a Tree Node $N$ in an exploration tree $\mathcal{T}$ is defined as a set of the following attributes:

**State** ($s_t$): Represents the accumulated historical context from the initiation of the process up to the current time step $t$, encapsulating all preceding reasoning paths and actions. Formally, the state at time $t$ is given by

$$s_t = \{u, a_1, o_1, \ldots, a_{t-1}, o_{t-1}\}, \quad (6)$$

including the task description $u$ and history at step $t$.

**Action** ($a_t$): denotes the specific operation performed at the current node, which affects the subsequent state. The action is selected by the policy language agent $\pi$ and is conditioned on the current state and reasoning path.

**Reward** ($r_t$): the immediate feedback received from environment after performing action $a_t$. In most language agent tasks, the immediate rewards from environments are set to zero or very sparse. For example, WebShop (Yao et al., 2022) only provides a final reward from 0 to 1 at the end of trajectories.

**Children** ($\mathcal{C}$): is represented by a list containing nodes explored at the next step.

**Q-value** ($q$): represents the expected total future reward achievable starting from the current state $s_t$, taking action $a_t$. The Q-values are updated once an exploration tree is constructed. We will introduce how we extract Q-values in the following section.

### 4.2.2. TREE CONSTRUCTION

We construct the reasoning trees in Algorithm 2. With each step in a trajectory formalized as a TreeNode, the entire trajectory is a branch within an exploration tree. To explicitly construct an exploration tree that captures potential generations from the root node (i.e., the initial task), exploring new trajectories can be viewed as expanding new branches from the existing TreeNodes. For any non-leaf tree node, effective generation can be achieved by: **1)** directly exploring and adding new child nodes that differ from the existing ones. **2)** For each branch that reaches a leaf node, we assess its quality based on the final reward provided by the environment. If the branch yields a zero reward, we stop generation on that branch's nodes, thereby reducing ineffective generation.

We introduce our strategy for tree construction in detail as follows.

**Tree Pruning.** In practice, we have found that the average depths of tree searching for agent tasks are large. Building an exploration tree and expanding every potential tree nodes may lead to heavy cost to the trajectory generation. To address this, we propose several strategies to reduce the computational burden during tree construction. We employ

---

**Algorithm 2** Constructing a Reasoning Tree

---

**Input**: A LLM agent $\pi_\theta$, a given task description $u$, a trajectory $\tau_0$ from the training set $\mathcal{D}_{expert}$ on task $u$, max exploration depth $D$, max exploration width $W$

Initialize a root node $U$ with state $s \leftarrow u$, depth $t \leftarrow 0$, reward $r \leftarrow 0$, action $\leftarrow$ *null*, children set $\mathcal{C} \leftarrow \{\}$

Initialize the reasoning tree $\mathcal{T}$ with $U$

The expansion node queue $E \leftarrow [u]$

**Subroutine** ProcessExpansionQueue($E$):

**while** $E$ is not empty **do**

   Get a node $N \leftarrow E$.pop with state $N.s$, action $N.a$, reward $N.r$, children set $\mathcal{C}$ at step $N.t$

   **if** the number of children in $N.\mathcal{C} < W$ and $N.t \le D$ **then**

      Sample a new trajectory $\tau$ based on state $N.s$

      Get a new branch $b$ constructed on $\tau$ and merge $b$ in node $N.\mathcal{C}$

      **if** $\tau$ achieves a non-zero final reward **then**

         Push all the nodes on $b$ with $N.t \le D$ into $E$

      **end if**

   **end if**

**end while**

**End Subroutine** ProcessExpansionQueue

Call ProcessExpansionQueue($E$)

Construct a branch $b$ with $\tau_0$ and merge in $U.\mathcal{C}$

Push all the nodes on $b$ with depth $t$ and $t \le D$ into $E$

Call ProcessExpansionQueue($E$)

**return** the reasoning tree $\mathcal{T}$

---

pre-pruning techniques to lower the generation costs when constructing an exploration tree for each task. First, we limit the expansion of tree nodes to the early stages of a trajectory (e.g., the first three to five steps, depending on the environment's complexity, with details provided in Appendix A.2).

Next, when a branch leads to a zero-outcome reward at its leaf node, we propagate a `Stop expansion` signal from the leaf node back to the earliest unexpanded intermediate node on that branch. This helps prioritize the generation of optimal trajectories given a limited generation budget. This construction process is illustrated in Figure 2.

With a set of exploration trees, we aim to efficiently gather effective step-wise signals for training an effective process reward model. Since most language agent tasks only return an outcome reward at the end of the trajectory, which is stored at the leaf nodes of the exploration tree, we need to develop methods to leverage these outcome rewards to generate effective intermediate signals.

**Extracting Q-values.** After constructing an exploration tree, with the outcome rewards stored in leaf node rewards, we estimate the Q-values for each intermediate node leveraging

$$Q(s_t, a_t) = r_t + \gamma \max_{a_{t+1} \sim \mathcal{C}_t} [Q(s_{t+1}, a_{t+1})], \quad (7)$$

where $\gamma$ is the discount factor, $s_{t+1}$ is the new state after action $a_t$, $\mathcal{C}_t$ is the children set containing nodes explored at the next step, and the expectation is over actions $a_{t+1}$ drawn

from the policy $\pi$. We provide the pseudocode of Q-value estimation on the exploration trees in Appendix A.3.1.

### 4.3. QNet Training

Inspired by the value function representing the expected long-term value in Q-learning (Watkins & Dayan, 1992), we extract Q-values for each node on the exploration trees using Equation 7. For each node $N = (s, a, q, \dots)$ in the collected exploration trees, we can then construct a supervised dataset $D_Q = \{(s, a, q)\}$ to train a Q-network (QNet), initialized from a supervised-fine-tuned LLM. Directly applying online Q-learning in language settings is often impractical, due to the unbounded action space of natural language and the sparse nature of rewards. Instead, by labeling each node with a corresponding Q-value offline and then training QNet in a purely supervised manner, we bypass the instability and excessive exploration costs that typical reinforcement learning loops would incur in high-dimensional language environments. The model architecture of QNet is introduced in Appendix A.2.2.

**Training Objective.** Given each exploration tree $\mathcal{T}$ with $n$ nodes: $\mathcal{T} = (N_1, N_2, \dots, N_n)$, we train the QNet $\mathcal{Q}_\phi$ by minimizing the Mean Squared Error (MSE) loss between the predicted Q-values $\hat{q}_t$ and the target Q-value $q$ calculated previously at each time step,

$$\mathcal{L}(\phi) = \frac{1}{n} \sum_{t=1}^{n} (\hat{q}_t - q_t)^2. \quad (8)$$

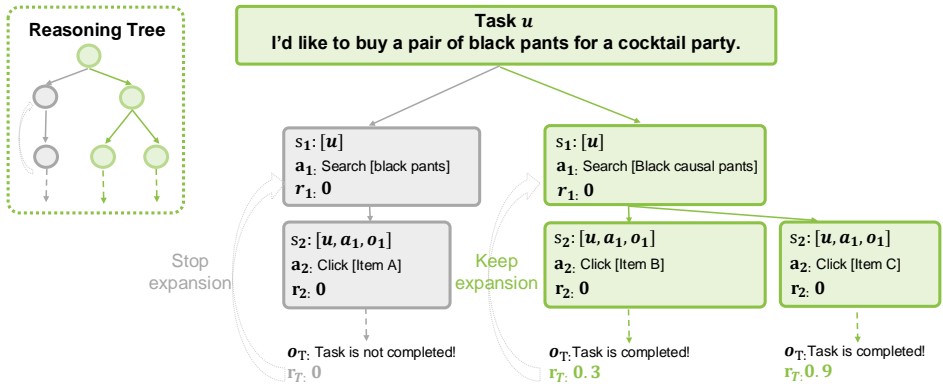

*Figure 2.* Illustrative example of constructing a exploration tree. Grey nodes represent the branches with a zero outcome reward. Once the leaf node with a zero outcome reward is detected, a `Stop expansion` signal will be sent back to the first unexpanded node on the branch. Green nodes are on branches where zero outcome reward is not detected and can keep expanding.

*Table 1.* The statistics of datasets (We follow the same setup as ETO (Song et al., 2024)). "Test-Seen" and "Test-Unseen" are test sets with seen and unseen cases respectively. "#Turns" means the average number of interaction turns for the SFT trajectories.

| Dataset | #Train | #Test-Seen | #Test-Unseen | #Turns |
|---------|--------|------------|--------------|--------|
| WebShop | 1,938 | 200 | - | 4.9 |
| SciWorld | 1,483 | 194 | 241 | 14.4 |
| ALFWorld | 3,321 | 140 | 134 | 10.1 |

By minimizing this loss, we encourage the QNet to produce consistent Q-value estimations across the sequence that align with the target Q-value $q$. This training objective emphasizes accurate Q-value predictions at each token, reinforcing the model's ability to assess the long-term value of actions throughout the trajectory.

### 4.4. Q-Guided Generation

The effectiveness of a good process reward model can be represented by whether it can lead to better agent self-improvement. Therefore, we conduct Q-guided generation for self-improvement to evaluate the effectiveness of QLASS. Q-guided generation enables agents to generate each step under the guidance of QNet. At each step, agents sample several actions and the one with the highest Q-value is executed by the agent. We provide a more detailed algorithm of Q-guided generation in Appendix A.3.2.

## 5. Experiment

In this section, we aim to evaluate the effectiveness of QLASS for solving complex agent tasks in the following aspects: 1) whether QLASS can aid better decision making on different complex agent tasks; 2) whether the Q-value in QLASS is an effective process reward to facilitate self-improvement; 3) whether QLASS can retain strong performance with reduced annotated data.

### 5.1. Setup

**Datasets.** We assess the ability of QLASS on Web-Shop (Yao et al., 2022), ALFWorld (Shridhar et al., 2021) and SciWorld (Wang et al., 2022a). These environments only provide a single outcome reward at the end of each trajectory. The statistics of three agent datasets are displayed in Table 1. The evaluation metric is the reward averaged on the test sets. During the sampling process, environments will give a termination signal when certain actions like "Click[Buy Now]" in Webshop are taken or the set maximum steps are reached. Details can be found in Appendix A.2.

**Training Setup.** In our work, we mainly use Llama-2-7B-Chat as base policy model and QNet backbone. We train our models mainly using 4 or 8 A6000 GPUs. The experiments on Webshop, including the training of SFT model, QNet, self-generation and Q-guided exploration, takes one or two days and the experiments on ALFWorld and SciWorld takes four or five days. The detailed hyper-parameters for training and model architectures can be found in Appendix A.2.

**Baselines**. 1) **SFT** (Chen et al., 2023) is the base agent after supervised fine-tuning on the expert data. 2) **RFT** (Rejection sampling Fine-Tuning) (Yuan et al., 2023) is a self-improvement baseline which is trained on the merged data consisting of successful trajectories sampled and expert data. 3) **ETO** (Song et al., 2024) is a self-improvement baseline which updates policy via constructing trajectory-level preference pairs and conducting DPO. 4) **PPO** (Proximal Policy Optimization) (Schulman et al., 2017): a reinforcement learning baseline which directly trains the base agents to optimize the final rewards. 5) **Best-of-N** samples N trajectories for each task and selects the one with the highest oracle outcome reward.

N is set to 6 in Table 2 and Table 3. All the inference-time baselines in the tables are under the same search budget for fair comparison. 6) **Closed-source agents**: GPT-

*Table 2.* Performance of all the baselines on WebShop, SciWorld and ALFWorld. The table is divided into two sections: the first presents the results of closed-source agents and the second includes open-sourced agents. ♠ indicates the baseline based on **GPT-4o**. In each column, the best result is **bolded** and the second-best result is underlined.[1]

| Method | WebShop | SciWorld | | ALFWorld | |
|---|---|---|---|---|---|
| | | **Seen** | **Unseen** | **Seen** | **Unseen** |
| GPT-4 | 63.2 | 64.8 | 64.4 | 42.9 | 38.1 |
| GPT-3.5-Turbo | 62.4 | 16.5 | 13.0 | 7.9 | 10.5 |
| Reflexion (Shinn et al., 2023)♠ | 64.2 | 60.3 | 64.4 | 45.7 | 55.2 |
| Base Agent (Llama-2-7B-Chat) | 17.9 | 3.8 | 3.1 | 0.0 | 0.0 |
| SFT | 63.1 | 67.4 | 53.0 | 60.0 | 67.2 |
| RFT (Yuan et al., 2023) | 63.6 | 71.6 | 54.3 | 62.9 | 66.4 |
| PPO (Schulman et al., 2017) | 64.2 | 59.4 | 51.7 | 22.1 | 29.1 |
| Best-of-N | 67.9 | 70.2 | 57.6 | 62.1 | 69.4 |
| ETO (Song et al., 2024) | 67.4 | 73.8 | 65.0 | 68.6 | 72.4 |
| MATH-SHEPHERD (Wang et al., 2023) | 66.9 | 72.5 | 63.3 | 72.9 | 76.9 |
| TS-LLM (Feng et al., 2023) | 67.7 | 73.7 | 65.3 | 74.3 | 79.1 |
| QLASS | **70.3** | **75.3** | **66.4** | **77.9** | **82.8** |

3.5-Turbo and GPT-4 with ReAct prompting (Yao et al., 2023), and methods depending on the emergent properties of self-reflection and planning from large proprietary models, and we use **Reflexion** (Shinn et al., 2023) as the baseline (use GPT-4o as the base model). We also include (7) **Process reward model based baselines**, including MATH-SHEPHERD (Wang et al., 2023) and TS-LLM (Feng et al., 2023).

**5.2. Evaluation Results**

In this section, we compare the performance of our QLASS with all the baselines on WebShop, SciWorld, and ALFWorld. We evaluate all algorithms using one-shot evaluation. The decoding temperatures are set to 0.7 for QLASS and Best-of-N and 0 for other baselines.

**Overall Baseline Comparison.** Results are summarized in Table 2. From Table 2, we can observe that QLASS consistently achieves the highest scores among all the open-sourced baselines, including both training-based methods and inference-based methods. QLASS also demonstrates comparable performance with the best proprietary baselines. Specifically, GPT-4 is the state-of-the-art model, but QLASS still outperforms it on all three benchmarks by 17.9% on average, especially on SciWorld and ALFWorld. Also, QLASS outperforms ETO and PPO consistently by over 5% on average, which are two strong baselines based on multiple stages of training, including supervised fintuning on expert trajectories, training reward models and doing DPO or PPO on the explored trajectories. We achieve better performance while avoiding the heavy cost (including the hyperparameter tuning on DPO / PPO).

**Inference-time Search Efficiency.** We compare QLASS and Best-of-N under different search budgets and visualize

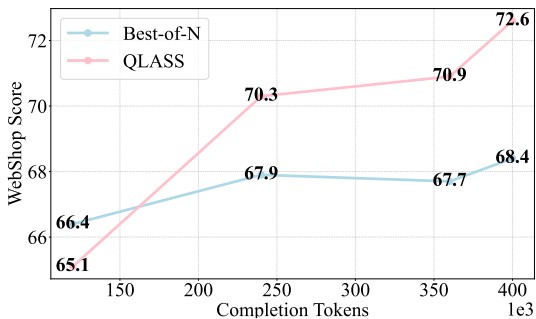

*Figure 3.* QLASS and Best-of-N under different search budgets. The x-axis represents the number of tokens consumed by the trajectories generated during inference averaged on all the tasks in each test set.

the results in Figure 3. We find that increasing the number of completion tokens will improve the performance of all inference methods. We can observe that QLASS is consistently better than Best-of-N under almost all the search budgets. Another notable observation is that compared with Best-of-N (68.4) under 400K tokens, QLASS (70.3) with only about half of search budgets under 240k tokens, outperforms the highest score of Best-of-N (68.4). Also, as the completion tokens approach 360K, Best-of-N begins to flatten, while the score of QLASS still gets improved by a relatively larger margin from 360K tokens to 400K tokens. This indicates that our approach is a more effective way to scale up the compute for inference-time self-improvement.

**Self-training Performance.** Since process reward modeling is an important module in our framework, we ablate on how different choices of process reward can affect the perfor-

---

[1]Part of the results results are adopted from (Song et al., 2024) and (Zhou et al., 2024).

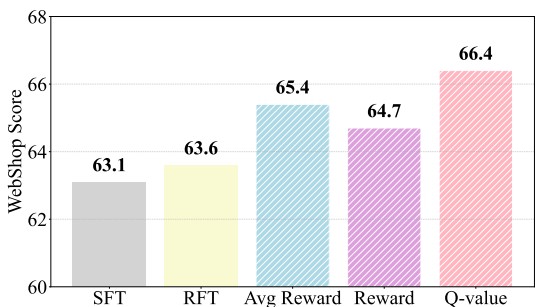

*Figure 4.* Self-training baselines. The three methods marked with diagonal stripes leverage different process reward modeling based on the same exploration trees constructed in Stage 2 to guide self-training data generation.

*Table 3.* Average reward comparison on WebShop with 1000 annotated trajectories for behavior cloning. The best result is **bolded**, and the second-best result is underlined.

| Method | WebShop | WebShop-1000 |
|---|---|---|
| SFT | 63.1 | 21.7 |
| ETO | 67.4 | 66.7 |
| Best-of-N | 67.9 | 47.1 |
| QLASS | **70.3** | **67.3** |

mance. We mainly experiment with three approaches of constructing process rewards for each intermediate nodes on the exploration trees: `Q-value` (ours) is to estimate Q-value for each state-action pair (i.e. each tree node except for root node) using Equation 7; `Avg reward` (Wang et al., 2023) computes the averaged the final rewards; `Reward` (Yuan et al., 2024) directly treats the final outcome reward and backpropagates it as the process reward for each intermediate step. In addition to the self-improvement at inference time, we also evaluate the effectiveness of QLASS for selecting high-quality data for self-training. We train the base agent on the SFT dataset in addition to the Q-guided generated data. Results are visualized in Figure 4. We observe that QLASS achieves the highest among all the self-training baselines, compared with RFT which leverages oracle outcome rewards to filter high-quality trajectories and baselines guided by other process reward models such as `Reward` and `Avg Reward`.

**Ablation on Process Reward Modeling.** We train three different process reward models guiding trajectory generation for self-training. Self-training results are in Figure 4. From Figure 4, we can observe that `Q-value` utilized by our QLASS yields the best performance, while the one using `Avg reward` is slightly better than the one directly using `Reward`, indicating the effectiveness of using `Q-value` to model process reward.

*Table 4.* The performance on a different base LLM on SciWorld.

| Base LLM | Method | SciWorld | |
|---|---|---|---|
| | | Seen | Unseen |
| Llama-2-13B | SFT | 68.1 | 57.6 |
| | ETO | 71.4 | 68.6 |
| | QLASS | 72.7 | 69.3 |

### 5.3. Fewer Annotations

In many real-world applications, collecting large amounts of expert-annotated data is both time-consuming and costly. To evaluate the effectiveness of our approach under such constraints, we designed this setup with fewer annotations to test how quickly the agents adapt to new environments in this section. We extract 1000 trajectories as a subset from the original 1938 trajectories. Under this setup, all baselines can only conduct behavior cloning with access to the SFT dataset of 1K trajectories. After that, baselines like RFT, ETO and QLASS which involve generation can explore on 1938 tasks. The performance comparison is listed in Table 3. We can observe that QLASS outperforms other methods trained on both the full WebShop training set and WebShop-1000 subset. This highlights the robustness of our method, especially its potential in scenarios with scarce expert data. While other methods like RFT and SFT show a significant drop in performance, QLASS remains effective, demonstrating the advantage of Q-guided generation for data selection even in annotation-limited environments.

### 5.4. Case Study

We pick out an example from ALFWorld in Figure 5 to showcase the difference between baselines and our models. The SFT agent correctly picks up the lettuce, cools it using the fridge, and places it on the countertop in the beginning. However, it continues performing redundant actions afterward, such as repeatedly opening and closing the fridge. The environment responds with "Nothing happened" until the agent exhausts its step limit, failing to recognize the task is already complete. By contrast, the QLASS uses a stepwise reward mechanism. Once it has cooled the lettuce and placed it on a countertop, it gains no further reward from reopening the fridge or replacing the lettuce. Consequently, the QLASS avoids futile actions and terminates as soon as the goal is satisfied, successfully completing the task in fewer steps. We can observe that Q-value gradually grows with the number of steps, but suddenly converges to an extremely high or low value at Action 5, indicating it is a key step that differentiates success and failure, where QLASS assigns "cool lettuce with fridge 1" with very high Q-value, only gives 0.10 to "close fridge 1" that leads to nonsense behavior.

**Goal:** put a cool lettuce on a countertop.

| | |
|---|---|
| **Action1: go to countertop 3**      **SFT** | **Action1: go to countertop 3 (q=0.25)**   **QLASS** |
| *Observation:* sees lettuce 1, among other items. | *Observation:* sees lettuce 1. |
| **Action2: take lettuce 1 from countertop 3** | **Action2: take lettuce 1 from countertop 3 (q=0.41)** |
| *Observation:* picks up lettuce 1. | *Observation:* picks up lettuce 1. |
| **Action3: go to fridge 1** | **Action3: go to fridge 1 (q=0.65)** |
| *Observation:* fridge 1 is closed. | *Observation:* fridge 1 is closed. |
| **Action4: open fridge 1** | **Action4: open fridge 1 (q=0.71)** |
| *Observation:* fridge is now open. | *Observation:* fridge is now open. |
| **Action5: close fridge 1** | **Action5: cool lettuce 1 with fridge 1 (q=0.89)** |
| *Observation:* you closed fridge 1. | *Observation:* "You cool the lettuce 1 using the fridge 1." |
| **Action6: open fridge 1** | **Action6: go to countertop 1 (q=0.95)** |
| *Observation:* fridge is now open. | *Observation:* sees a butterknife 1. |
| **Action7: close fridge 1** | **Action7: put lettuce 1 on countertop 1** |
| *Observation:* you closed fridge 1. | *Observation:* "You put the lettuce 1 on countertop 1." |
| *Repeatedly until hit max of turns* | *Task Completed Successfully.* |

**Action5**
close fridge 1 (q=0.10)
cool lettuce 1 with fridge 1 (q=0.89)

*Figure 5.* One example on the ALFWorld, the right is QLASS and the left is the SFT baseline.

## 5.5. Ablations Across Different Base Policy Models

To validate the robustness of our method across different model architectures, we also conduct experiments on a large base model: Llama-2-13B. As shown in Table 4, QLASS still outperforms the baseline reported in Song et al. (2024) on the SciWorld benchmark.

## 6. Conclusion

In this paper, we introduce QLASS, a novel approach that enhances open-source language agents at inference time by integrating Q-value-based process guidance. By modeling the Q-value at each intermediate step during planning, our method offers step-wise feedback that surpasses the limitations of outcome-based reward models, particularly in complex, long-horizon tasks. Through extensive experiments, we have demonstrated that QLASS significantly improves the language agent to search more intelligently. Moreover, our method demonstrates strong performance even in scenarios with limited annotated data used for behavior cloning. This work paves the way for more efficient and scalable self-improvement techniques in language models, enabling them to tackle complex tasks with reduced reliance on human annotations.

## Impact Statement

Our work aims to improve the inference-time search for open language agents by learning Q-based process reward which reduces the reliance on extensive human annotations. By leveraging self-generated data and more efficient step-wise evaluations, we hope to lower the barrier to developing advanced language agents for tasks where data is scarce. This has the potential to broaden access to high-quality AI-driven solutions, enabling researchers, even those with limited resources—to deploy models that can handle complex, multi-turn tasks without much human annotation efforts.

From an ethical and societal standpoint, more capable and autonomous language agents raise considerations regarding misinformation, biases, and potential misuse. In particular, any method that enhances an agent's ability to explore and generate content at scale should be paired with careful oversight and robust filtering mechanisms. While our approach focuses on improving performance through better trajectory evaluation rather than content manipulation, we acknowledge that improved capabilities can be leveraged for harmful or deceptive purposes if not properly governed.

## Acknowledgement

This work is supported by DARPA ANSR program FA8750-23-2-0004, DARPA ECOLE Program HR00112390060, ONR grant N00014-23-1-2780, NSF 2211557, NSF 2119643, NSF 2303037, NSF 2312501, SRC JUMP 2.0 Center, Amazon Research Awards, and Snapchat Gifts.

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

# A. Appendix

## A.1. Discussion

**Why supervised train the offline QNet using LLM as the backbone instead of directly using deep Q-learning?** Directly applying deep Q-learning (Watkins & Dayan, 1992) to language agents face critical challenges. First, the action space (all possible text outputs) is unbounded and orders of magnitude larger than typical RL environments (e.g., Atari's 18 discrete actions). Standard exploration strategies like $\epsilon$-greedy fail because random text sampling rarely yields meaningful rewards or trajectories. Second, language tasks often involve sparse rewards, destabilizing Q-learning's reliance on frequent reward signals. Pure online Q-learning would suffer from high gradient variance and require infeasible exploration budgets.

Initializing the value function model from a well-pretrained large foundation model can encode rich linguistic and reasoning priors, as well as world commonsense knowledge (Bansal et al., 2024; Song et al., 2024; Xu et al., 2022; Wang et al., 2023). So we initialize our QNet with the LLM trained in the agent environment to embrace both knowledge during pretraining and agent specific capabilities, thus boosting the adaption to the long-term value modeling.

## A.2. Experimental details

### A.2.1. DATASETS

We follow the setup of ETO (Song et al., 2024) to use the three agent tasks for our experiments.

(a) WebShop is an online shopping environment. The available action types for agents include *search[keywords]* and *click[value]*. The agent is instructed to complete the task with ReAct(Yao et al., 2023)-style response. The instruction is specified in Figure 6.

(b) ALFWorld (Shridhar et al., 2021) consists of interactive TextWorld environments paralleling the embodied worlds. In this setup, agents must explore and complete complex household tasks. The ALFWorld dataset includes both seen and unseen evaluation sets. The seen set tests in-distribution generalization, while the unseen set evaluates out-of-distribution generalization, featuring entirely new task instances for the agents to solve.

(c) SciWorld (Wang et al., 2022a) is a text-based virtual platform designed around conducting basic scientific experiments across ten task categories, such as thermodynamics and electrical circuits. Agents engage in embodied, interactive environments to grasp scientific concepts through practical tasks. Each task in ScienceWorld includes optional subgoals, with the final reward calculated based on the achievement of these subgoals.

We have summarize the statistics of SFT datasets for behavior cloning on all the environments in the main body. Note that the default reward from the environment is zero for the intermediate step before terminal. For self-generation and tree construction, we set the maximum step as 5 in WebShop and 18 in ALFWorld and SciWorld. For inference, we set the maximum step as 5 in WebShop and 40 in ALFWorld and SciWorld. The instruction templates are displayed in Figure 6, 7 and 8.

### A.2.2. QNET

**Model Architecture.** Our QNet is designed by sharing the backbone of the Large Language Model (LLM) and appending a value head to predict Q-values. Specifically, we utilize a pre-trained LLM, denoted as $LLM_\theta$, which serves as the foundational model for encoding input sequences. The value head is a Multi-Layer Perceptron (MLP) that takes the hidden states from the LLM and outputs scalar Q-value predictions.

Formally, given an input sequence of tokens $\mathbf{x} = (x_1, x_2, \ldots, x_n)$ representing a state-action pair, the LLM produces hidden states $\mathbf{h} = (h_1, h_2, \ldots, h_n)$:

$$\mathbf{h} = LLM_\theta(\mathbf{x}), \tag{9}$$

where $h_t \in \mathbb{R}^d$ represents the hidden state at time step $t$, and $d$ is the hidden size of the LLM.

The value head $MLP_\phi$ processes each hidden state $h_t$ to predict the corresponding Q-value $\hat{q}_t$:

$$\hat{q}_t = MLP_\phi(h_t), \tag{10}$$

---

**Algorithm 3** Q-value Estimation

---

**Input**: A reasoning tree $\mathcal{T}$ with a root node $U$, discount factor $\gamma$

**Procedure**   Update_Q_Values($N$)
**if** $N.\mathcal{C} = \varnothing$ **then**
    **return**                                                         $\triangleright$Leaf nodes do not update
**end if**
**for** node $N_{\text{child}}$ in $N.\mathcal{C}$ **do**
    Update_Q_Values($N_{\text{child}}$)                                    $\triangleright$Recursively update child nodes first
**end for**
$N.q = N.r + \gamma \max_{N_{\text{child}} \in N.\mathcal{C}} (N_{\text{child}}.q)$             $\triangleright$Update Q-value after all children are updated
**End Procedure**

Update_Q_Values($U$)                                $\triangleright$Start the update process from the root
$Q_{\min} = \min_{N \in \mathcal{T}}(N.q)$
$Q_{\max} = \max_{N \in \mathcal{T}}(N.q)$
**for** node $N$ in $\mathcal{T}$ **do**
    $N.q = \frac{N.q - Q_{\min}}{Q_{\max} - Q_{\min}}$                                   $\triangleright$Apply min-max normalization
**end for**
**return** the reasoning tree $\mathcal{T}$ with estimated Q-value of each node

---

where $\hat{q}_t \in \mathbb{R}$ is the predicted Q-value at time step $t$, and $\phi$ denotes the parameters of the MLP.

The MLP consists of multiple layers with ReLU activations, culminating in a linear layer that outputs a scalar Q-value. This design allows the model to capture complex patterns in the hidden representations and map them to accurate Q-value estimates.

**Training Objective.** Given an explored trajectory $\mathbf{x} = (x_1, x_2, \ldots, x_n)$ with an associated target Q-value $q$, we train the QNet by minimizing the Mean Squared Error (MSE) loss between the predicted Q-values $\hat{q}_t$ and the target Q-value $q$ at each time step:

$$\mathcal{L}(\theta, \phi) = \frac{1}{n} \sum_{t=1}^{n} (\hat{q}_t - q)^2. \tag{11}$$

By minimizing this loss, we encourage the QNet to produce consistent Q-value estimations across the sequence that align with the target Q-value $q$. This training objective emphasizes accurate Q-value predictions at each token, reinforcing the model's ability to assess the long-term value of actions throughout the trajectory.

**Implementation Details.** In practice, we implement the value head as an MLP with two hidden layers of size 1024 and ReLU activation functions.

The entire model, including the LLM and the value head, operates in bfloat16 precision to optimize memory usage without sacrificing performance. The LLM backbone remains frozen or fine-tuned depending on the specific experimental setup, allowing us to leverage pre-trained language representations while focusing on learning accurate Q-value predictions through the value head. By integrating the value head with the LLM, our QNet effectively combines language understanding with reinforcement learning principles, enabling the agent to make informed decisions based on both linguistic context and estimated future rewards.

### A.3. Algorithms

#### A.3.1. PSEUDOCODE OF Q-VALUE DISTILLATION

In this section, we provide the pseudocode of how we distill the Q-value from an exploration tree in Algorithm 3

---

**Algorithm 4** Q-guided Generation

---

**Input**: A LLM agent $\pi_\theta$, a given task description $u$, an action set $\mathcal{A}_t$ containing $M$ candidates at step $t$, a trained QNet $\mathcal{Q}_\phi$, sampled trajectory number $N$, max trajectory length $L$
traj_candidates = [ ]
**for** $i = 1$ **to** $N$ **do**
   Initialize state $s_i \leftarrow [u]$
   **for** $t = 1$ **to** $L$ **do**
      Collect a set of action candidates $\mathcal{A}_t \leftarrow$ Sample $a \sim \pi_\theta(a \mid s_i)$ for $M$ times
      $a_t \leftarrow \text{argmax}_{a \sim \mathcal{A}_t} \mathcal{Q}_\phi(s_i, a)$                           ▷Select the best action with max Q-value
      Take action $a_t$, and receive new observation $o_t$ from environment
      $s_i \leftarrow s_i + [a_t, o_t]$                ▷Update state with executed action and new observation
      **if** $s_i$ is the final state **then**
         **break**                                ▷Exit loop if stop condition is met
      **end if**
   **end for**
   traj_candidates.append($s_i$)
**end for**
Select the best trajectory $s$ with best final reward $s$.reward from traj_candidates

---

### A.3.2. Q-GUIDED GENERATION

In this section, we present the pseudocode of Q-guided generation in Algorithm 4, which is a critical component of our framework.

**Perturbation augmented generation**. In WebShop, due to the limited diversity of sampled actions, we introduce augmenting action diversity with perturbation during this stage, which is realized by prompting GPT-3.5-Turbo to paraphrase the task description. This utilization of perturbation enables us to inject more variability into the prompts that guide action selection, substantially enriching the range and relevance of possible actions. Such enhanced prompts help prepare the model to handle more diverse and unforeseen situations effectively. We augmented the action diversity in all inference-based algorithms when evaluating in WebShop for fair comparison. Noted that it costs too much on ALFWorld and SciWorld, so we only conduct perturbation on the WebShop.

We introduce our implementation details and examples as follows. We use GPT-3.5-Turbo to perturb the task descriptions using the prompt *"Paraphrase the text: {task description}"*. We show an illustrative example on a WebShop task in Figure 9.

### A.4. Hyper-parameters

We summarize the hyper-parameters used across both all stages of QLASS in this section. The hyper-parameters leveraged in behavior cloning and self-training is in Table 5. Training QNet shares all the same hyperparameters, except that the number of training epochs is set to 2.

*Table 5.* Hyperparameters used in QLASS.

| Hyperparameter | Value |
|---|---|
| Batch size | 64 |
| Number of training epochs | 3 |
| Weight decay | 0.0 |
| Warmup ratio | 0.03 |
| Learning rate | 1e-5 |
| LR scheduler type | Cosine |
| Logging steps | 5 |
| Model max length | 4096 |
| Discount factor $\gamma$ | 0.9 |
| Maximum expansion depth $D$ on WebShop | 3 |
| Maximum expansion depth $D$ on SciWorld | 6 |
| Maximum expansion depth $D$ on ALFWorld | 8 |
| Action candidate set size $M$ for inference | 2 |
| Sampled trajectory number $N$ for self-training | 1 |
| Exploration temperature | 0.7 |

---

**WebShop Instruction**

You are web shopping.
I will give you instructions about what to do.
You have to follow the instructions.
Every round I will give you an observation and a list of available actions, you have to respond an action based on the state and instruction.
You can use search action if search is available.
You can click one of the buttons in clickables.
An action should be of the following structure:
**search[keywords]**
**click[value]**
If the action is not valid, perform nothing.
Keywords in search are up to you, but the value in click must be a value in the list of available actions.
Remember that your keywords in search should be carefully designed.
Your response should use the following format:
**Thought: I think ...**
**Action: click[something]**

*Figure 6.* The instruction prompt provided to language agent on WebShop.

## SciWorld Instruction

You are a helpful assistant to do some scientific experiment in an environment.
In the environment, there are several rooms: kitchen, foundry, workshop, bathroom, outside, living room, bedroom, greenhouse, art studio, hallway
You should explore the environment and find the items you need to complete the experiment.
You can teleport to any room in one step.
All containers in the environment have already been opened, you can directly get items from the containers.

The available actions are:
open OBJ: open a container
close OBJ: close a container
activate OBJ: activate a device
deactivate OBJ: deactivate a device
connect OBJ to OBJ: connect electrical components
disconnect OBJ: disconnect electrical components
use OBJ [on OBJ]: use a device/item
look around: describe the current room
examine OBJ: describe an object in detail
look at: describe a container's contents
read OBJ: read a note or book
move OBJ to OBJ: move an object to a container
pick up OBJ: move an object to the inventory
pour OBJ into OBJ: pour a liquid into a container
mix OBJ: chemically mix a container
teleport to LOC: teleport to a specific room
focus on OBJ: signal intent on a task object
wait: task no action for 10 steps
wait1: task no action for a step
---
Now, it's your turn and here is the task.
Task Description:
Your task is to boil lead. For compounds without a boiling point, combusting the substance is also acceptable. First, focus on the substance. Then, take actions that will cause it to change its state of matter.

Figure 7. The instruction prompt provided to language agent on SciWorld.

## ALFWorld Instruction

Interact with a household to solve a task. Imagine you are an intelligent agent in a household environment and your target is to perform actions to complete the task goal. At the beginning of your interactions, you will be given the detailed description of the current environment and your goal to accomplish.
For each of your turn, you will be given the observation of the last turn. You should first think about the current condition and plan for your future actions, And then output your action in this turn. Your output must strictly follow this format:"Thought: your thoughts.\nAction: your next action".

The available actions are:
1. go to {recep}
2. task {obj} from {recep}
3. put {obj} in/on {recep}
4. open {recep}
5. close {recep}
6. toggle {obj} {recep}
7. clean {obj} with {recep}
8. heat {obj} with {recep}
9. cool {obj} with {recep}
where {obj} and {recep} correspond to objects and receptacles.
After your each turn, the environment will give you immediate feedback based on which you plan your next few steps. if the environment output "Nothing happened", that means the previous action is invalid and you should try more options.

Your response should use the following format:

Thought: <your thoughts>
Action: <your next action>
Now, it's your turn and here is the task.

You are in the middle of a room. Looking quickly around you, you see a cabinet 6, a cabinet 5, a cabinet 4, a cabinet 3, a cabinet 2, a cabinet 1, a coffeemachine 1, a countertop 3, a countertop 2, a countertop 1, a drawer 3, a drawer 2, a drawer 1, a fridge 1, a garbagecan 1, a microwave 1, a shelf 3, a shelf 2, a shelf 1, a sinkbasin 1, a stoveburner 4, a stoveburner 3, a stoveburner 2, a stoveburner 1, and a toaster 1.
Your task is to: put a clean knife in countertop.

Figure 8. The instruction prompt provided to language agent on ALFWorld.

**Perturbation**

Paraphrase the task: I need a long lasting 6.76 fl oz bottle of l'eau d'issey, and price lower than 100.00 dollars

I'm looking for a 6.76 fl oz bottle of L'eau D'Issey that lasts a long time and costs less than $100.

I need a durable 6.76 ounce bottle of L'eau D'Issey, with a price under $100.

I am searching for a long-lasting 6.76 fluid ounce L'eau D'Issey perfume for less than 100 dollars.

*Figure 9.* An illustrative example on task perturbation.

