# OpenReview forum: "QLASS: Boosting Language Agent Inference via Q-Guided Stepwise Search"
_ICML.cc/2025/Conference — ICML 2025 poster_

### Official Review · Reviewer_agwr · 2025-03-09

**Overall Recommendation:** 3

**Summary:**

Given the limitation that current agent task do not possess high-quality granular reward signals, this work proposes the QLASS (Q-guided Language Agent Stepwise Search) method to automatically explore states , learn step-wise values, and apply these value-based heuristics are inference time. QLASS is shown effective in improving downstream agent performance through efficient value learning procedures.

**Claims And Evidence:**

1. The main claim that learning and applying Q-value to language agents improves performance, is supported by the main result in Table 2, where the proposed QLASS method achieves the highest result on all three benchmarks on varied settings.
2. The claim on inference-time search efficiency in Figure 3, may not be fully supported due to incomplete computation cost calculation. IIUC, the baseline Best-of-N method only requires inference-time scaling, therefore the cost (“Completion Tokens”) calculated in Figure 3 are sufficient; nonetheless, for the QLASS method, additional training process on policy and reward models are required, yet Figure 3 only computes the inference-time cost. A more detailed discussion on the overall computation cost could be helpful. Furthermore, the metric “Completion Tokens” is not very intuitive to understand, e.g., if 150 tokens is sufficient for generating 1 or multiple responses (which is sufficient for common agent inference), or what’s the expected value/distribution of “Completion Tokens” on the tested benchmarks. Knowing this information would help decide which are the critical regions on Figure 3 x-axis, and if QLASS is more empirically useful than the simple Best-of-N method.
3. For ablation study in Section 5.5, while QLASS still achieves higher results than other methods with the 13B model. However, comparing to 7B model results in Table 2, which should be presumably lower since smaller models are usually weaker, QLASS with 13B model actually underperforms QLASS with 7B model. More justification on this inferior result, or further more comprehensive studies on model scaling could be helpful.

**Essential References Not Discussed:**

The related work section has discussed the relevant literature rather comprehensively. However, there is one paper that I find relevant but not cited in this paper:
Koh, Jing Yu, et al. "Tree search for language model agents." arXiv preprint arXiv:2407.01476 (2024).

**Experimental Designs Or Analyses:**

1. Corroborated by the “Methods and Evaluation Criteria” section, additional experiments using varying N examples for BC could be helpful, especially the case when N=0.
2. The choice of main model, LLama-2-7B-chat, is somewhat unexpected. As more upgraded versions of similar-sized Llama models (3.1, 3.2) have been released, it is a bit unclear why the somewhat older Llama-2 model is selected. Similar confusion holds for choosing the “-Chat” version specifically.

**Methods And Evaluation Criteria:**

The method design is well-motivated and reasonable in general. However, one question that I have is the necessity of the “Behavioral Cloning” stage introduced in Section 4.1. Because it is a warm-start for the language agent and not related to the core Q-value model, it seems that the QLASS method could potentially work without this BC process. While BC is a beneficial process given that the main task-solving agent experimented in this work is open-source models, another experiment that (1) do not involve this BC stage, or (2) ablates more Ns in section 5.3 (including N=0) could be helpful. Furthermore, using closed API models (e.g., gpt-4o, claude) as the task-solving agent (while keeping the QNet training and Q-Guided training with open-source models), could offer more information on the effectiveness of the designed modules.

**Other Comments Or Suggestions:**

For Figure 4, the bar plot shows Q-value achieves 66.4, while Table 2 shows 70.3. What causes the difference between?

**Other Strengths And Weaknesses:**

The paper is written with clarity; all results are presented clearly in tables and figures.

**Questions For Authors:**

The method (specifically BC and Q-learning) requires supervision data corresponding to the test set, I wonder how the method could be generalized to agent tasks without compatible training examples or ground-truth environmental rewards?

**Relation To Broader Scientific Literature:**

The key Q-learning idea of this paper is related to reinforcement learning and process reward modeling. The results are consistent with several reward modeling works in that PRMs are effective (more so that ORMs) in improving multi-step agentic tasks.

**Theoretical Claims:**

The paper introduces the Q-value learning process in symbolic expressions, which reads reasonable.

---

> ### Author Rebuttal · Authors · 2025-04-01
>
> Dear Reviewer agwr,
>
> We greatly appreciate your insightful comments for our work. Here are our responses to your questions.
>
> > 1 Incomplete computation cost calculation and explanation on “Completion tokens”
>
> We would like to clarify that all the inference-time methods in Figure 3, i.e., Best-of-N and QLASS, share the same policy training computation. The computation overhead of QLASS lies in exploring trajectories to build the reasoning tree and then train the reward model. Second, the training cost is once while inference cost is almost infinite from a long-term perspective. The terminology is also used in prior work[1].
>
> For the computation leveraged for exploration, we have included the hyperparameters in Table 5 and discussed in Appendix A.2, where we kindly direct the reviewer for further details.
>
> The “Completion tokens” in Figure 3 x-axis refers to all the inference cost leveraged in Best-of-N and QLASS. A rough estimation is that each complete trajectory includes 40 * 1e3 tokens. Take 400 * 1e3 on the x-axis as an example, it means the inference cost includes 10 generated trajectories.
>
> > 2 Justification of QLASS with 13B model weaker than 7B
>
> We summarize and compare the results on 7B and 13B models in the table below.
>
> | | SciWorld-seen | SciWorld-unseen |
> |-|-|-|
> |SFT-7B|67.4|53.0|
> |SFT-13B|68.1 (+0.7) |57.6 (+4.6)|
> |ETO-7B|73.8 | 65.0|
> |ETO-13B|71.4 (-2.4)| 68.6 (+3.6) |
> |QLASS-7B|75.3| 66.4  |
> |QLASS-13B|72.7 (-1.6) | 69.3 (+2.9)|
>
> On the seen set, both the ETO-13B and QLASS-13B models show slightly lower performance compared to their 7B counterparts (-2.4 and -1.6, respectively). This might suggest some degree of overfitting in the larger 13B models, where the model may have specialized too much on the training data.
>
> On the unseen set, all 13B models show significant improvements over their 7B counterparts (+4.6, +3.6, and +2.9, respectively). This indicates that the larger 13B models demonstrate better generalization capabilities, performing better on new, unseen data.
>
> > 3 The necessity of BC stage and the impact of BC with different N examples
>
> We observe that both the effectiveness of exploration and the QNet depend heavily on the initial capabilities of the model. Recent research [2] suggests that a good initialization of the LLM achieved through SFT is crucial for reducing the search space and is beneficial for inference-time scaling. We empirically found that with N=0, the quality of exploration was very poor, making it difficult to train a good reward model. Some tasks show nearly zero performance w/o BC as shown in Table 2. In practical applications, it is generally feasible to secure a small set of high-quality training trajectories annotated by experts or researchers. But scaling up the dataset is difficult due to cost, time, and maintaining consistency in data quality.
>
> We added experiments on the setup of using 200 examples for behavior cloning and summarize the results of leveraging different N examples for BC in the table below.
>
> |  |WebShop| WebShop-1000 | WebShop-200 |
> |-|-|-|-|
> | SFT | 63.1|21.7| 20.9 |
> | ETO | 67.4 |66.7| 52.1 |
> | BoN|67.9| 47.1| 45.1|
> | QLASS|70.3| 67.3| 53.7|
>
> We can observe from the table that QLASS consistently outperforms other baselines under setups leveraging different BC examples, demonstrating its robustness across different BC datasets.
>
> > 4 Choice of Llama-2-Chat
>
> We chose the Chat version because it is more appropriate and more easily adapted to the multi-turn, interactive agent tasks that our paper focused on. We chose Llama-2-Chat for our experiments because we are building on the code from ETO [3], which also uses Llama-2-Chat. Additionally, FastChat that we use for serving and generation, provides stable support specifically for Llama-2 models. Due to resource limitations, we are unable to rerun all experiments on Llama-3 at this stage but plan to include Llama-3 results in the future.
>
> > 5 More API-based models
>
> We have GPT4, GPT3.5-Turbo, and GPT-4o based Reflexion included in our main Table 2. We also add additional experiments on GPT4o on the WebShop. We kindly direct you to the discussion in Q3 of Reviewer hiva.
>
>
> > 6 Explanation of 66.4 of Q-value in Figure 4 and 70.3 of QLASS in Table 2
>
> The experimental setups are different. As we stated in L358-361, results in Figure 4 are self-training baselines where QNet is used for self-training data generation. QLASS in Table 2 is leveraged to provide inference-time guidance.
>
> > 7 Missing citation
>
> Thanks for bringing this missing citation to us. We will add it in our revised version.
>
> We hope that our answers can resolve your concerns.
>
> [1] Zhang, D., Zhoubian, S., Hu, Z., et al. (2024). Rest-MCTS: LLM self-training via process reward guided tree search.
>
> [2] Team, K., Du, A., Gao, B., et al. (2025). Kimi K1.5: Scaling reinforcement learning with LLMs.
>
> [3] Song, Y., Yin, D., Yue, X., et al. (2024). Trial and Error: Exploration-Based Trajectory Optimization for LLM Agents.

---

### Official Review · Reviewer_hiva · 2025-03-13

**Overall Recommendation:** 3

**Summary:**

This paper proposes QLASS, a method for Q-value estimation in process reward modeling, providing stepwise guidance for language agents. QLASS consists of four main stages: SFT to train the LLM agent, exploration tree construction, QNet training and Q-guided generation. Compared to multiple baselines, QLASS achieves significant performance improvements across various tasks with fewer training data, demonstrating its efficiency and effectiveness.

**Claims And Evidence:**

The paper claims that QLASS enhances LLM-based agents' decision-making performance in complex interactive tasks and enables self-improvement under limited supervision. These claims are well-supported by experimental results, which show that QLASS outperforms several baselines across multiple benchmarks.

**Essential References Not Discussed:**

No.

**Experimental Designs Or Analyses:**

The paper provides a comprehensive comparison with various agent training paradigms, effectively demonstrating QLASS's advantages. However, the absence of comparisons with other process reward modeling methods weakens the argument that QLASS is the best approach for process reward estimation.

**Methods And Evaluation Criteria:**

QLASS utilizes Q-value-based stepwise guidance to optimize agent behavior, which is a well-justified approach. The construction of QNet appears to be a key advantage of this method compared to others, however, the evaluation does not assess QNet in combination with multiple different LLM-based agents, which makes it difficult to fully demonstrate the superiority of the approach.

**Other Comments Or Suggestions:**

The visual consistency of figures could be improved to enhance clarity and readability. Unifying the style across diagrams would make the presentation more polished.

**Other Strengths And Weaknesses:**

Strengths

1. QLASS introduces an innovative stepwise search strategy, effectively addressing the issue of suboptimal decision-making caused by outcome-based rewards.

2. The paper conducts thorough experiments across multiple complex interactive environments, demonstrating QLASS's effectiveness and robustness.

3. QLASS maintains strong performance even with reduced annotated data, making it valuable for real-world applications where labeled data are scarce.

Weaknesses

1. QNet's effectiveness is not tested with multiple LLM-based agents, making it unclear whether its benefits extend beyond the specific agent used in the experiments.

2. The paper does not compare QLASS with alternative process reward modeling techniques.

3. Lack of cross-domain generalization experiments.

**Questions For Authors:**

1. I am curious about the generalization ability of QLASS. If the model is trained with SFT and QNet only on WebShop data, can it still achieve performance improvements on the ALFWorld test set?

2. Since QNet is designed by sharing the backbone of the LLM, I would like to know whether the inherent performance of the backbone affects the quality of Q-value predictions.

3. Could you demonstrate the effectiveness of Q-guided generation by applying the trained QNet to other LLM agents?

4. How does a QNet constructed using QLASS compare to other process reward models (e.g., process rewards built using the Math Shepherd paradigm or process rewards derived from advanced closed-source models) in terms of performance when used with the same LLM agent?

**Relation To Broader Scientific Literature:**

The key contributions of this paper build upon and extend multiple areas of prior research, including LLM-based agent reasoning and process reward modeling.

**Theoretical Claims:**

The paper employs Q-learning for process reward modeling, which is theoretically sound.

---

> ### Author Rebuttal · Authors · 2025-04-01
>
> Dear Reviewer hiva,
>
> We sincerely thank you for the constructive suggestions and positive feedback. We will address your concerns below.
>
> > W1: Lack of experiments of applying QLASS to different LLM agents
>
> We understand that the reviewer encourages us to experiment on diverse LLM agents to validate the effectiveness of QLASS. In Section 5.5, we experimented with base agents with different model sizes to investigate the effectiveness of QLASS on different LLM agents. Additionally, we experimented on applying QLASS on GPT-4o (stated in Q3 below). We also summarize the results below.
>
> | Base LLM | Method | SciWorld-seen | SciWorld-unseen |
> |----------------|--------|---------------|-----------------|
> | Llama-2-7B     | SFT    | 67.4          | 53.0            |
> | Llama-2-7B     | ETO    | 73.8          | 65.0            |
> | Llama-2-7B     | **QLASS**  | **75.3**          | **66.4**            |
> | Llama-2-13B    | SFT    | 68.1          | 57.6            |
> | Llama-2-13B    | ETO    | 71.4          | 68.6            |
> | Llama-2-13B    | **QLASS**  | **72.7**          | **69.3**            |
>
> From the table above, we can see that QLASS can effectively enhance the performance significantly across diverse LLM agents.
>
>
> > W2: Lack of comparison with other process reward modeling methods
>
> To validate the effectiveness of QLASS compared with other PRM, we compared our Q-value based method with Avg reward[2] and Reward[3]. Avg reward computes the averaged final rewards; Reward directly treats the final outcome reward and backpropagates it as the process reward for each intermediate step. In addition to the self-improvement at inference time. More details are included in Section 5.2, L370-384. We leverage Q-value, Avg reward, Reward to guide agent self-training.
>
> Results are in the table below.
> | Q-value | Avg reward | Reward |
> |-------------|-------------|-------------|
> | 66.4 | 65.4 | 64.7 |
>
> We can observe that Q-value achieves the highest among all the PRM baselines, demonstrating the effectiveness of QLASS compared with other process reward models.
>
> > W3: Lack of cross-domain generalization experiments.
>
> In our current setup, we experimented on unseen test sets on SciWorld and ALFWorld in Table 2, which is a commonly adopted setup to test the out-of-distribution generalization ability of LLM agents in prior works[1].
>
> Also, we respectfully clarify that the tasks and action spaces explored in our experiments, such as WebShop (a shopping task) and ALFWorld (a navigation task), involve distinctly different knowledge bases with minimal overlap. Consequently, it is impractical to apply the QNet trained on WebShop to enhance performance on ALFWorld due to these fundamental differences. We acknowledge the importance of this clarification and will include this point in the revised version of our paper.
>
> > Q1: The cross-domain generalization ability of QLASS
>
> We have discussed this question in W3 and we kindly direct the reviewer to the discussion in W3.
>
> > Q2: How the inherent performance of the backbone affects the quality of Q-value predictions
>
> In practice, we found that QNet, when initialized by sharing the backbone and weights from the agent's LLM, performs slightly better than when trained from scratch.
> We previously employed llama3.2 3B as the base model for both the agent model and QNet. However, llama3.2 3B performs poorly in our multi-turn, interactive agent tasks. It also struggled to provide effective process rewards during inference time. These shortcomings may stem from the model's capacity limitations, as the 3B size may be inadequate for handling the complexity required for solving these specific tasks.
>
>
> > Q3: Applying QNet to different LLM agents
>
> We have experiments showing that our method can also work on 13B LLM in Table4. Also,
> we add additional experiments by applying the trained 7B QNet to GPT4o on the Webshop. Noted that GPT-4o is not specifically trained on agent tasks so the 7B results after behavior cloning can perform better.
>
> | Method | Performance|
> |-------------|-------------|
> | GPT-4o | 54.5 |
> | GPT-4o w/ QLASS | 56.2 |
>
> > Q4: Comparison with other process reward models
>
> We include detailed discussion on comparison with other PRMs in W2.
>
> We hope that our answers can resolve your concerns.
>
>
> [1] Song, Y., Yin, D., Yue, X., et al. (2024). Trial and Error: Exploration-Based Trajectory Optimization for LLM Agents.
>
>
> [2] Wang, P., Li, L., Shao, Z., et al. (2023). Math-shepherd: A label-free step-by-step verifier for LLMs in mathematical reasoning.
>
>
> [3] Yuan, L., Li, W., Chen, H., et al. (2024). Free process rewards without process labels.

---

### Official Review · Reviewer_dLz3 · 2025-04-02

**Overall Recommendation:** 3

**Summary:**

The paper proposes an LLM self-improvement recipe for tasks where there is a (possibly sparse) external verification signal, inspired by the Q-learning algorithm for Markov Decision Processes. Experiments conducted on three domains (ALFWorld, SciWorld and WebShop) show that the proposed recipe can yield performance improvements for LLM-as-agents, and scales well with increased inference-time compute budgets.

[Post rebuttal update] The additional experiments and details submitted by the authors in their rebuttal address most of my questions.

**Claims And Evidence:**

The main claims are:
1. The proposed approach for constructing reasoning trees and computing Q-values can produce a good quality dataset for LLM self-improvement.
2. Using predicted Q-values to guide LLM generation can yield a good policy for agent tasks.
3. The general QLASS pipeline produces good performance at low cost for agent tasks, relative to other baselines.

The evidence for (1) can be substantially improved with some additional analysis:
- For a given cost budget (w.r.t. number of tokens or LLM calls or latency) there is a tradeoff between trying a task for longer (i.e. higher T; trajectory length) vs. backing off and exploring other nodes as in Algorithm 2. By carefully varying T, D (reasoning tree max_depth) and W (reasoning tree max_branching) we can empirically understand this tradeoff.
- The paper conjectures that outcome reward models yield inferior policy learning results than process reward models because sometimes the resulting policy may be inefficient. An ablation experiment with varying gamma (discount factor) would be a great way to verify or falsify this conjecture -- does gamma=1 allow inefficient policies and as gamma < 1 the learned policies become more timestep-efficient?
- Can we still produce good Q-value datasets after multiple steps of LLM self-improvement? An experiment where Stages 2 and 3 of Algorithm 1 are re-iterated a few times would shed light on this question.

The evidence for (2) seems adequate. Figure 3 can be made more convincing by including other inference-compute scaling techniques.

The evidence for (3) is missing some important baselines: (even allowing for excluding the ones discussed in Appendix A.1).
For instance, from the paper's citations & related work -- process reward models learned via random rollouts (e.g. Uesato'22, Lightman'23, Wang'23, Chen'24); learned outcome reward models (Snell'24, Wang'24a, Shinn'23); MCTS approaches to building the "reasoning tree" (e.g. TS-LLM Feng'23, ReST-MCTS* Zhang'24). A representative approach from each of these 3 threads of research are important baselines to compare.

**Essential References Not Discussed:**

N.A.

**Experimental Designs Or Analyses:**

How were the hyper-parameters for QLASS (e.g. expansion depth D) selected across the different task domains? Was it by monitoring performance on the test set? If so, there may be unfair bias against the other tested baselines. This is an important missing detail that should be clarified for a rigorous experimental setup.

**Methods And Evaluation Criteria:**

The benchmarks used (WebShop, Alfworld, SciWorld) are well-motivated. Evaluation metrics such as task success rate and token counts are also reasonable.

Evaluating the costs of the proposed approach vs. baselines needs clarification or perhaps a more rigorous approach.
QLASS has a dataset generation step, followed by Q-Net training, and followed by using Q-Net during inference.
The only costs reported are in Figure 3 which only reports the cost of Q-Net during inference.
The other costs are important to contextualize.

**Other Comments Or Suggestions:**

Section 3: Mention the MDP problem setting that the Q-learning algorithm is designed for (because it is not yet apparent that the tasks tackled later in the paper are modeled well by an MDP).

Notation of N has a conflict. N is the number of expert trajectories during SFT; and also denotes a node in the exploration tree.

Appendix A.2.2. Notation clash between (x1 ... xn) representing tokens being input to the LLM vs. the subscript representing the time-step of the task trajectory (each step of the task trajectory will have a sequence of tokens right?)

Section 4.4: the second paragraph is superfluous and can be cut. Algorithm 2 should be in the main paper instead.

Algorithm 2: "Get a new branch b constructed on \tau" this is not described adequately in the paper. When a trajectory is sampled from a state, is each call to the LLM made into a node? And the state corresponding to the node is the concatenation of all previous inputs and outputs of the LLM from the root node to that node?
Line numbers are not printed, so "repeat function in Line 5-12" is hard to interpret; perhaps refactor into an Algo block and a Subroutine block.

Main paper Section 5.1 should mention the perturbation augmentation done for WebShop. Why was the cost for that step high on SciWorld/AlfWorld (since it was just about paraphrasing the task descriptions)?

Algorithm 3: When describing Q-value estimation in the main paper, mention that the Q-values are normalized to be in [0,1].

**Other Strengths And Weaknesses:**

The biggest strength of the paper is the very strong empirical performance of QLASS across the three tested domains.

**Questions For Authors:**

Please see questions in the other review responses above.

**Relation To Broader Scientific Literature:**

The paper adequately describes the related works on self-improvement recipes for LLMs, using LLMs as agents, and using process reward models to derive fine-tuning reward signals on intermediate steps of agent tasks.

**Theoretical Claims:**

The paper does not make any theoretical claims.

---

### Decision · Program_Chairs · 2025-05-01

**Decision:**

Accept (poster)

**Comment:**

The paper proposes a technique for LLM self-improvement that is inspired by the Q-learning algorithm in RL. The reviewers agreed that the problem studied in the paper is timely and well-motivated (improving LLM-powered AI agents), the proposed technique is interesting (generating tree-structured rollouts and estimating Q-values from sparse outcome reward signals) and the empirical evidence is sound and convincing.

In their rebuttal, the authors submitted several additional experiments that addressed all of the concerns and suggestions raised by the reviewers. Including all of these additional experiments substantially strengthens the paper.